DOI: 10.1038/s41467-018-06357-0　**OPEN**

# Evidence for magmatic carbon bias in $^{14}$C dating of the Taupo and other major eruptions

Richard N. Holdaway[1,2], Brendan Duffy [3] & Ben Kennedy [4]

Prehistoric timescales, volcanic hazard assessment, and understanding of volcanogenic climate events rely on accurate dating of prehistoric eruptions. Most late Quaternary eruptions are dated by $^{14}$C measurements on material from close to the volcano that may be contaminated by geologic-sourced infinite-age carbon. Here we show that $^{14}$C ages for the Taupo (New Zealand) First Millennium eruption are geographically arrayed, with oldest ages closer to the vent. The current eruption wiggle match date of 232 ± 5 years CE is amongst the oldest. We present evidence that the older, vent-proximal $^{14}$C ages were biased by magmatic $CO_2$ degassed from groundwater, and that the Taupo eruption occurred decades to two centuries after 232 CE. Our reinterpretation implies that ages for other proximally-dated, unobserved, eruptions may also be too old. Plateauing or declining tree ring cellulose $\delta^{13}$C and $\Delta^{14}$C values near a volcano indicate magmatic influence and may allow forecasting of super-eruptions.

[1] Palaecol Research Ltd, Hornby, P.O. Box 1659, Christchurch 8042, New Zealand. [2] School of Biological Sciences, University of Canterbury, Private Bag, Christchurch 4800, New Zealand. [3] School of Earth Sciences, University of Melbourne, Melbourne 3010, Australia. [4] Department of Geological Sciences, University of Canterbury, Private Bag, Christchurch 4800, New Zealand. Correspondence and requests for materials should be addressed to R.N.H. (email: turnagra@gmail.com)

Radiometric dating of pre- and early historic volcanic eruptions based on tephra-chronologic control of terrestrial, ice-, lake-, and ocean-core stratigraphy underpins assessment of volcanic hazards[1] and volcanogenic climate change[2,3]. Contamination of samples by carbon containing non-equilibrium levels of [14]C can distort radiocarbon ages[4,5]. The use of inaccurate radiometric eruption dates, particularly for events that occurred remote from contemporary literate societies, can bias a chronology and affect interpretations of data keyed to that chronology, from archaeology[6,7] to volcano-climatic connections[2,3]. Users rarely, however, question the basis for a chronology. A recent example is the anchoring in 2011 of the SP04 (South Pole) ice core chronology[8] on the 1980-vintage, 186 CE date[9] for the Taupo (New Zealand) First Millennium eruption (hereafter, the Taupo eruption, or "Taupo"), rather than the 1995-vintage, 232 ± 15 years CE date[10]. The Taupo eruption has a currently accepted date of 232 ± 5 years CE[11] and we challenge this date here.

The most recent vent of the Taupo supervolcano lies in the northeast of the 616 km$^2$ caldera Lake Taupo in the central North Island, New Zealand. The lake and caldera are within the central, 125 km long, rhyolite-dominated segment of the Taupo Volcanic Zone (TVZ)[12], a ~2 Myr old, actively rifting centre of silicic magmatism. The TVZ hosts a 4.2 GW geothermal system that is fed from shallow (2–7 km deep) heat sources[12]. The region

receives about 1200 mm annual rainfall, which recharges aquifers developed in variably permeable pyroclastic deposits that are increasingly fractured with age/depth and interspersed with lacustrine or paleosol aquitards[13]. More than 33% of the region's water wells are < 20 m deep, demonstrating the proximity of the water table to the ground surface.

The Taupo eruption has been the subject of one of the most intensive[10,11,14], and earliest[15], uses of radiocarbon for dating prehistoric eruptions. The first wiggle match [14]C date[10] followed >40 [14]C ages with stratigraphic relationship to one or other of the component tephras of the eruption series[16] (Fig. 1). A second, 25-sample, wiggle match date[11] on a different tree from the same forest also burnt and buried within the final ignimbrite[17] of the Taupo eruption gave a date (232 ± 5 years CE) indistinguishable from the first. The agreement between the results of the two dates, led to the rejection of previously-measured ages, many of which were geologically younger than the wiggle match dates[11].

Here we evaluate the pre-wiggle match [14]C ages for the Taupo eruption and reveal a relationship between increasing distance from the vent and decreasing [14]C age of samples associated with the eruption. We interpret δ[13]C values of the wiggle match tree samples as indicating contamination by magmatic $CO_2$ injected into the regional groundwater by accumulating basaltic magma beneath the rhyolitic magma chamber. Similar plateauing was observed in records for other eruptions. We suggest that the

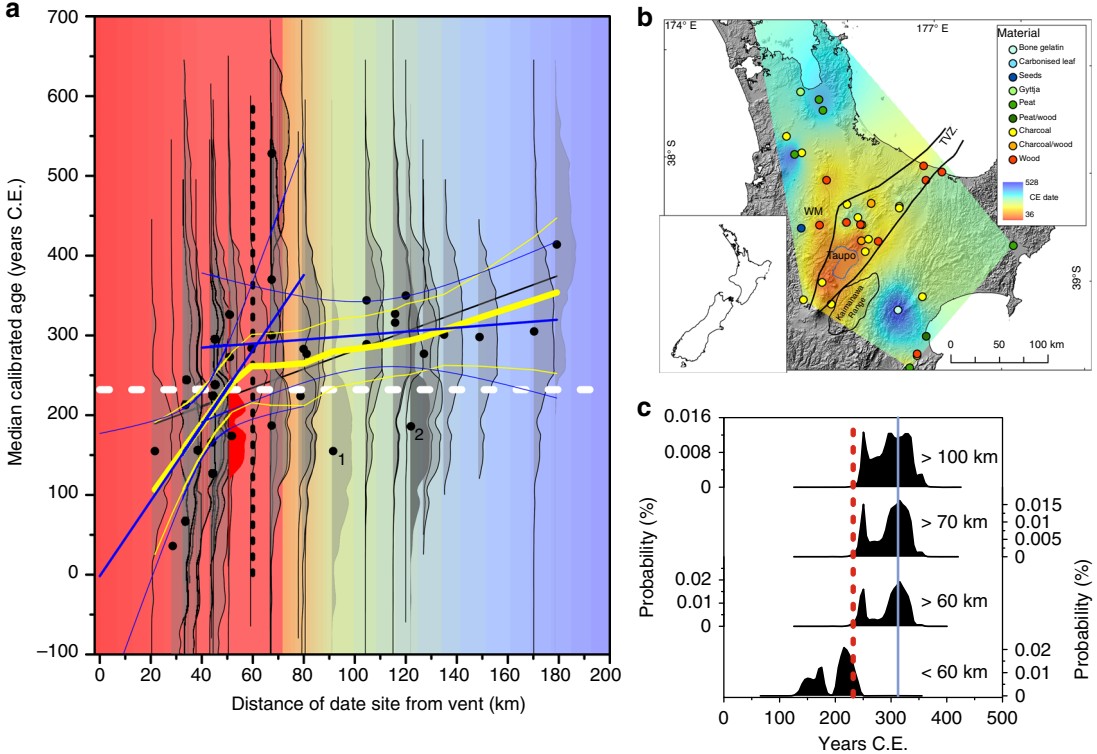

**Fig. 1** Radiocarbon ages for the Taupo First Millennium eruption in relation to distance and direction from the eruptive vent. Ages are oldest nearest the vent ("Taupo") and youngest away from the Taupo Volcanic Zone (Data in Supplementary Information Table 1). **a** Relationship between median calibrated age and distance from the presumed vent: yellow, local regression, 0.6 smoothing factor; blue, linear regressions for sample median ages <60 km and >60 km from vent, with 95% confidence limits; vertical broken line, limit of linear relationship between distance and age; white broken line, date of eruption from second wiggle match analysis; red, oldest; blue, youngest. Outer wood wiggle match age distribution in red. Note monotonic distributions of ages on samples >60 km from the vent, with 90+% of their distributions younger than the wiggle match age. 1, NZ165, on material from Arapuni, adjacent to the Waikato River; 2, NZ1059, on peat from Lake Poukawa, probability distribution (darker shading) extends well into the calibrated range of the monotonic distant ages. **b** Geographic pattern of magnitude of median calibrated radiocarbon ages: red, oldest; blue, youngest. Legend shows age gradient and dated materials. **c** Summed probability distributions of calibrated radiocarbon ages for the Taupo eruption, for dates other than the two wiggle match series, for pooled samples at different distances from the eruption vent. Red broken line, second wiggle match date for eruption; blue line, highest probability of combined calibrated ages on samples >60 km

plateauing of both $\Delta^{14}C$ and $\delta^{13}C$ values in tree rings could be useful in forecasting major eruptions.

## Results

**Radiocarbon ages versus distance from vent.** We re-examined the corpus of $^{14}C$ ages for the Taupo eruption, and found a highly significant relationship between the median calibrated SHCal13[18] calendar ages and distance from the assumed vent position[17], for sites within 60 km from the vent (Fig. 1a). Furthermore, if the material type is considered, the wood and charcoal data sets that contain sufficient samples for age/distance relationship to be evaluated both display the same pattern of younging with increasing distance from the vent as in the aggregate data set (Supplementary Table 1). The median age for the final rings of the second wiggle match tree was amongst the oldest. The median ages are geographically as well as linearly arrayed (Fig. 1b; Supplementary Fig. 1), precluding laboratory error as a basis for differences from the wiggle match ages[11]. A combined calibrated date (OxCal 4.3, 'combination' option) for the non-wiggle match $^{14}C$ ages of $254 \pm 34$ CE (median 241 CE) overlapped both wiggle match ages at 1 sigma; and the accompanying probability distribution had a second peak after 300 CE (Fig. 1c).

The peak probabilities for the combination of ages on material <60 km from the vent were in the 170 s and 210 s CE (Fig. 1c), whereas the bimodal probability distribution of the combined ages on material >60 km from the vent had a small older peak c. 250 CE and a much higher peak at c. 310 CE. Combining dates from successively greater distances (60, 70, 100 km) from the vent did not move the overall distribution but increased the size of the later (younger) peak.

The distribution of median values for 50 random radiocarbon ages based on the wiggle match age of 232 CE, with measurement errors set at 25 years (similar to those of the wiggle match series), did not resemble the data range in Fig. 1a, producing only about half the spread of ages and none in the range of the young ages distant from the vent (Supplementary Fig. 2).

Many of the pre-wiggle match radiocarbon measurements were on wood or charcoal and were measured before the possibility of inbuilt age of wood or charcoal (when the sample was from non-peripheral wood or small twigs) was fully taken into account. A possible explanation for old ages is the coincidence that most of the proximal charcoal and wood ages had significant inbuilt ages and that wood and charcoal ages on samples from >60 km from the vent were on short life wood. Inbuilt age bias cannot apply to a wiggle match age so we offer a more likely and mechanistically supported explanation for this geographic array.

**Magmatic $CO_2$ in regional groundwater.** The radial geographic pattern centred on the volcano (Fig. 1) suggests that volcanic processes, such as the injection[19] of magmatic $CO_2$ containing only 'infinite age' or 'old' carbon (no residual $^{14}C$ content) have introduced a geographic age bias. Radiocarbon measurements on dissolved inorganic carbon (DIC) in Lake Taupo water, and on plants and animals[20] in the lake food web, yielded ages of c. 4000 years on zero-age materials. The ages and anomalously high carbon stable isotope ratios ($\delta^{13}C$) for the DIC, reveal significant (20–42%, Fig. 2) levels of contamination by magmatic carbon (Fig. 2). In addition, radiocarbon ages[20] measured on bone protein of a brown rat (*Rattus norvegicus*, NZA12025, $2139 \pm 55$ years BP) and of a small duck (New Zealand scaup, *Aythya novaeseelandiae*, NZA11209, $2674 \pm 65$ years BP), both collected in 2000 CE at the Lake Taupo shoreline, show that high levels of magmatic carbon still enter the local food web. New Zealand scaup feed on aquatic plants and invertebrates[21,22] and brown rats are omnivores, swim well, and in shoreline habitats, as at

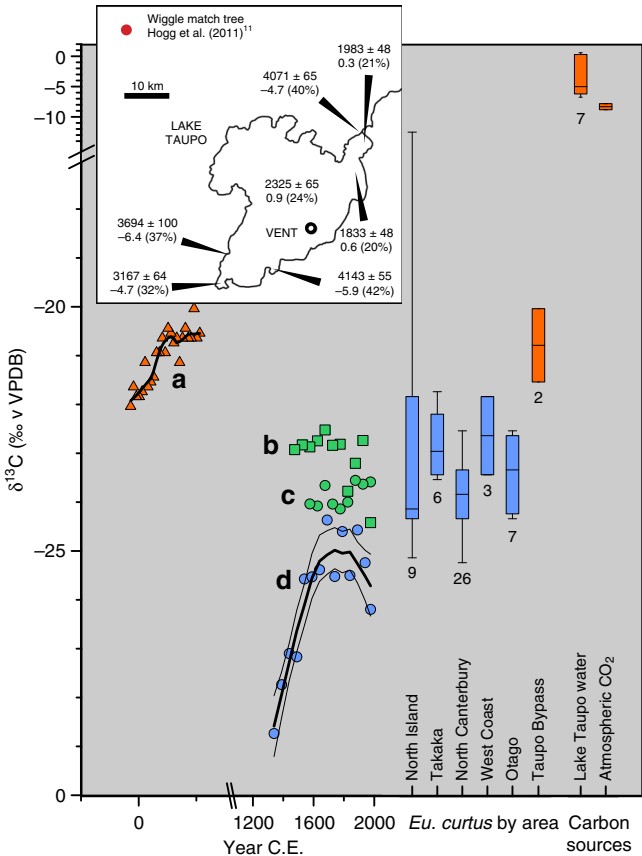

**Fig. 2** Comparisons between carbon stable isotopic ratios ($\delta^{13}C$) of the wiggle match tree rings with other trees at various distances from the eruption vent (see Fig. 1 for colour code), of bone gelatin of a species of herbivorous bird (Dinornithiformes: Emeidae: *Euryapteryx curtus*), and for Taupo lake water. Values of $\delta^{13}C$ in trunk wood cellulose and in the other organisms indicate the availability of magmatic ($^{14}C$-free) carbon dioxide for photosynthetic uptake by trees. **a** Wiggle-match tree[11], **b**, **c** two matai (Podocarpaceae: *Prumnopitys taxifolia*) trees growing in the open at 'Glenore', Hawkes Bay, 68 km east of vent, **d** rimu (Podocarpaceae: *Dacrydium cupressinum*) tree, Fox River, Paparoa National Park, West Coast, South Island. Box-whisker plots (maximum, minimum, median, lower and upper quartiles; numbers, sample sizes): bone gelatin $\delta^{13}C$ for *Euryapteryx curtus* (from areas within (red, Taupo Bypass, <60 km from vent) and (blue) remote from TVZ; Lake Taupo DIC[20], and current atmospheric $CO_2$. Note axis breaks. Inset: Lake Taupo: conventional radiocarbon ages of DIC in lake water (with measurement s.d., $\delta^{13}C$ values, and percentage of infinite age carbon contaminating dated sample)[20]

Lake Taupo, they obtain food by diving[23]. Therefore, both the duck and rat were obtaining their carbon from the Lake Taupo aquatic food web, and their $\delta^{13}C$ values reflect contributions of geologic $CO_2$ to that food web. Another rat, whose $^{14}C$ age (NZA12024, $-261 \pm 60$ BP)[20], was not offset, was part of the terrestrial food web.

Deep groundwaters in the TVZ contain a magmatic $CO_2$ component of $6 \pm 2\%$ to $14 \pm 5\%$[24]. Surface outflow from Lake Taupo potentially perturbs $^{14}C$ ages of samples downstream from Lake Taupo[20], as it does downstream of Long Valley Caldera in California[25], but the magmatic $CO_2$ must contaminate the groundwater throughout the volcanic zone, providing a secondary source of $CO_2$ for vegetation[26]. Even when diluted by recharge from other catchments, the groundwater source of $^{14}C$-depleted $CO_2$ would bias radiocarbon dates in the Waikato valley

beyond the Taupo volcanic zone[20]. The wiggle match tree was at 51.8 km from the vent (Supplementary Information Table 1) near the 60-km limit of contamination posited here, but it was only c. 30 km from Lake Taupo and <20 km from the Waikato River. The groundwater is continuous with both the lake and the river so a contamination offset should be present though not necessarily large.

**$\delta^{13}C$ as an indicator of magmatic $CO_2$ contamination.** Potential non-magmatic drivers of differences between $\delta^{13}C$ of leaves and wood, and between pre-industrial and present wood include the Suess Effect, fractionation between foliage and wood, and taxon differences. The Suess Effect has been factored into the values plotted in Fig. 2. Wood $\delta^{13}C$ values are typically ≤2 ‰ higher than those of leaves[27]. Typical $\delta^{13}C$ values of tree wood and leaves outside volcanic influences range from −32 to −24 ‰[28], increasing as they grow into the canopy and gain access to the tropospheric reservoir $CO_2$ (Fig. 2d). In general, vegetation growing beneath dense canopies, even of grass, exhibit low $\delta^{13}C$ values in comparison to those for isolated plants or leaves growing within the canopy[29–32]. Subcanopy vegetation includes saplings that will possibly mature into canopy trees over a period of decades to a few centuries, so the $\delta^{13}C$ values of these saplings, retained in the inner wood of the mature tree, could be up to 10‰ lower than those of the later rings of the mature tree, even with the leaf-wood fractionation. A plot of tree height or ring number against $\delta^{13}C$ should be a 'hockey-stick' curve, as is evident in the plot for the *Dacrydium* in Fig. 2. The wiggle match tree $\delta^{13}C$ values depart from this pattern, both in the lack of a substantial tail of low $\delta^{13}C$ values in its inner wood and in its anomalously high $\delta^{13}C$ values throughout.

Inter-taxic differences in $\delta^{13}C$ for C3 plant foliage such as New Zealand forest trees can reach ~3‰ but are minor in relation to those resulting from their positions within closed canopy forests and conditions of growth: all taxa show the same canopy values of foliage $\delta^{13}C$[28]. Values in wood can reach −22.5 ‰ in isolated trees away from volcanic centres (Fig. 2), but both wiggle match trees[10,11] grew in dense forest in a swampy valley[33]. This environment provided an ideal situation for uptake of respiratory[5] and magmatic $CO_2$ by roots[26] and (degassed from the soil) subcanopy leaves before it mixed with tropospheric reservoir $CO_2$. The $\delta^{13}C$ values (−22 to −20 ‰) of the second wiggle match tree[11] were significantly higher than those of New Zealand forest trees (Fig. 2) and lacked the lower range of values in the earliest decades of growth expected for a subcanopy sapling, supporting the view that the tree incorporated magmatic $CO_2$ degassed from local groundwater and taken up through the roots[34,35], as well as from the subcanopy air[5,30–32,36,37].

**Moa bone gelatin $\delta^{13}C$ as proxy for magmatic $CO_2$ contamination.** The bone protein $\delta^{13}C$ values (Fig. 2) of four female moa (*Euryapteryx curtus*) provide indirect support for the influence of groundwater on the level of degassing of old carbon. Soon after post-eruptive revegetation of the Taupo region, five of these large (c. 50 kg), now extinct, herbivorous birds were entombed in a cavity formed by a phreatic explosion immediately post-eruption[38] or eroded later in the ignimbrite c. 1.6 km from the present lake shore and 20 km from the vent (Supplementary Fig. 3). One moa yielded a radiocarbon age (NZA34021, 2026 ± 35 conventional [14]C years BP; median calibrated date 5 years CE), two centuries older than 232 CE, which is impossible given that the bird's deposition post-dated the eruption and had to have post-dated the recovery of the vegetation (Supplementary Figs. 3–5). The next oldest (NZA34021, 1617 ± 35 years BP, median calibrated date 485 CE) could have occupied the new vegetation

at that date, but it was still 250 years older than the other two birds dated from the deposit (Supplementary Figs. 4, 5).

The minimum offset of 300 years in NZA34021 would require that the sample included at least 3.5% [14]C-free carbon. Successive combinations of the radiocarbon ages maintained the date ranges for the three youngest ages, but constraining the oldest moa age to post-eruption violated the combinatorial and phase algorithms (Supplementary Fig. 4) confirming that the oldest moa age, and probably the next, must have been biased by [14]C-free carbon in the bird's diet. Their bone protein $\delta^{13}C$ values, which were consistent with those of the second wiggle match tree and significantly higher than values reported for the same moa species elsewhere in New Zealand (Fig. 2; Supplementary Fig. 5), imply contamination by [14]C-free carbon in the environment close to the vent, with continued outgassing from the lake and its environs. The comparative $\delta^{13}C$ data set encompasses glacial to interglacial climates, from 30 ka to the species' extinction in the early 15th century CE[39].

Other causes are usually invoked for anomalously old [14]C ages, where there is a definite date for the organism's death. For example, a linear relationship between sample $\delta^{13}C$ and offset ages on bones of people killed at Herculaneum in 79 CE was attributed to a diet including marine fish. However, we suggest that these anomalous ages may also reflect the effect of magmatic $CO_2$ in the groundwater on the terrestrial components of the diet[40] because two [14]C ages on a sheep (*Ovis aries*) metatarsal from the same site, presumably unaffected by a marine diet, were also several decades too old[40]. The sheep and humans were consuming vegetation that grew on the flanks of Vesuvius[41] or within the Campi Flegrei caldera[42] and hence the [14]C ages of their bones contain a legacy from the $CO_2$ degassed from the magma bodies.

**Plateauing of $\Delta^{14}C$ and $\delta^{13}C$ in wiggle match series.** In addition to the $\delta^{13}C$ values of the Taupo second wiggle match samples being anomalously high[11] (Fig. 2), the $\delta^{13}C$ and $\Delta^{14}C$ values plateaued successively within the life of the tree (Figs. 2a, 3a, b). After c.135 years and c. 155 years, respectively, linear relationships with actual tree age broke down: the tree continued to grow but [14]C ages of the newly accreted wood were static (Fig. 3). The plateau in $\delta^{13}C$ values might be interpreted as recording the tree canopy reaching free air above the canopy, but this is improbable given that the $\delta^{13}C$ values were much higher than those for New Zealand forest subcanopy vegetation elsewhere[43] (Fig. 2), and that the $\delta^{13}C$ plateau was followed immediately (in the tree age) by the plateau in $\Delta^{14}C$. The $\delta^{13}C$ plateau probably reflects isotopic fractionation of $CO_2$ as it exsolved from the 'trigger' basalt beneath the rhyolite reservoir[44], and as it degassed from the groundwater.

Fractionation during degassing is governed by the $CO_2$ concentration in the groundwater[44]: as more magmatic $CO_2$ entered the groundwater, relatively more isotopically light $CO_2$ was degassed, reducing the $\delta^{13}C$ of the gas available for photosynthesis. The wood $\delta^{13}C$ recorded the net biospheric (and magmatic) $CO_2$ flux, and the $\Delta^{14}C$ the total $CO_2$ flux[4]. Residuals of both $\Delta^{14}C$ and $\delta^{13}C$ in relation to local regression fits exhibit quasi-periodic variation, which may be related to fluctuations in outgassing of magmatic $CO_2$ beneath the canopy. The period of 30–50 years excludes the possibility that the fluctuations are seasonal. We suggest that wiggle-matched radiocarbon measurements and associated $\delta^{13}C$ values should only be made on trees well outside a posited contamination area, which lived and died in the same time frame as the eruption of interest. This would be a fruitful area of future research and an independent test of our contamination hypothesis.

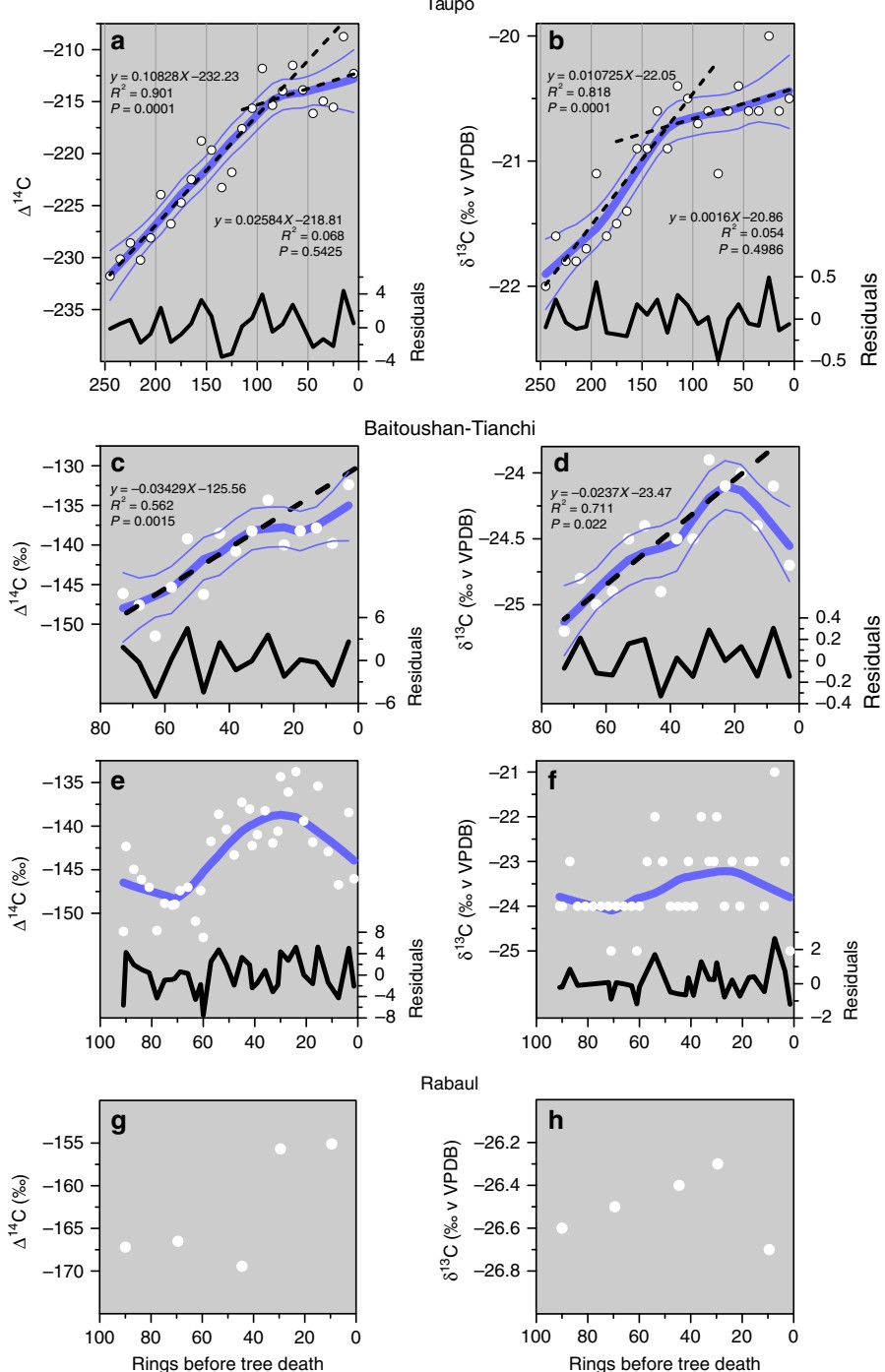

**Fig. 3** Patterns of variation in carbon isotopic ratios ($\Delta^{14}C$, $\delta^{13}C$) in trees close to the Taupo, Baitoushan-Tianchi, and Rabaul volcanoes before major eruptions. Carbon isotope ratios plateaued or declined as the eruptions approached. **a**, **b** Taupo ignimbrite (data, ref. [7]): thick blue line, 0.5 LOESS fit; thin blue lines, 2.5 and 97.5 percentiles of LOESS; broken black lines, linear fits before and after break in LOESS curve; solid black line residuals of LOESS fit. **c**, **d** Baitoushan-Tianchi (data, ref. [26]): details as in **a** and **b**. **e**. **f** Baitoushan-Tianchi (data, ref. [25]; $\delta^{13}C$ reported as integers); lines as in **a** and **b**, without linear fits. **g**, **h** Rabaul (data, ref. [27]): values only; insufficient data for LOESS fit

$CO_2$ respired from soil processes (decomposition and respiration) has been shown to bias $^{14}C$ ages[4,5], with 13–28% biospheric $CO_2$ recorded in the canopy air used in generating stem wood cellulose[4]. At night, $CO_2$ respired by the non-photosynthesising vegetation accumulates beneath a dense canopy[4,37], and as the highest net photosynthesis occurs in the morning[4], the respired $CO_2$ and any (both biospheric and magmatic) degassed from the soil contributes disproportionately to the stem wood. The effect is further enhanced by the cellulose of the lower part of the stem being produced from $CO_2$ taken up by leaves in the lower interior of the tree's canopy, where the air is sheltered from mixing with tropospheric $CO_2$[4]. Whereas the $^{14}C$ age bias attributable to biospheric $CO_2$ is limited by the residence time of soil organic material, significant quantities of magmatic $CO_2$ degassing into

the subcanopy air could add decades to millennia to the measured age of both plants and animals within the food web, as shown by the 'too-old' ages of the duck, rat, and two moa on the shores of Lake Taupo, and in vegetation growing within the Furnas caldera (Azores)[45].

Degassing of $CO_2$ from magma and carbonate rock has been reported from several regions and at different scales[46] and may have affected radiocarbon ages for eruptions. Several wiggle match dates[47,48] for a major eruption of Baitoushan-Tianchi volcano on the border of North Korea and China, securely dated by dendrochronology to 946 CE[49], are variably offset from the calibration curve depending on their location in the forest and distance from the crater. The best wiggle match age of 946 ± 3 CE, from 24 km distance, has a clear old offset from the calibration curve[50].

As in the Taupo tree, the $\delta^{13}C$ values for the wiggle match trees used to date this eruption ceased to increase over the 30 years before tree death (Fig. 3c–f), a feature also of the data for the Rabaul (Papua New Guinea) eruption[51] (Fig. 3g, h). The plateaux or declines in the $\delta^{13}C$ values mimic the Suess Effect apparent in the rimu tree growing on the West Coast of the South Island (Fig. 2d), but occurred well before the Industrial Revolution. The volcanic V-Seuss Effect in these wiggle match trees may reflect the different concentration and composition of Taupo (rhyolite/ basalt) source $CO_2$ compared with those of the Baitoushan-Tianchi and Rabaul eruptions (both dominantly dacite or andesite). Baitoushan-Tianchi inhabited a favourable section of the calibration curve, with pronounced age peaks and troughs at the eruption age. This highlighted the old offsets of wiggles from the calibration curve and led to unsubstantiated speculations about a magmatic carbon bias[50,52]. The Rabaul wiggle consisted of only 5 points, anchored by an ambiguous calibration. One age was questionably an outlier. As with all previous ages[53], they were obtained on wood that grew <10 km from the caldera rim, so the geographic contexts of the ages are not sufficiently different to show an age–distance relationship. An offset of ~135 years would account for the outlier. The three case studies suggest that more than 50% (and as much as 66%) of volcanic wiggle match ages display evidence for bias by magmatic carbon.

Two dates for the Taupo eruption unbiased by magmatic $CO_2$, were obtained from stratigraphically controlled samples bracketing the ignimbrite in the Hukanui Pool site[54,55], c. 68 km from the vent and separated from the Taupo Volcanic Zone by the northern Kaimanawa mountain range (c. 1200 m) (Fig. 1b). The first, for the boundary between eight ages (NZA7635, NZA7181, NZA7184, NZA10199, NZA8559, NZA8226, NZA6636, NZA7532)[54] on material below and within, and two (NZA7183, NZA7611)[54] above the ignimbrite was 360 ± 79 CE (Fig. 4), close to the peak probability for wiggle match age re-calculated with a uniform contamination by 5% old carbon applied to 25 ages (Fig. 4). A 5% level of contamination is well within that necessary to shift the $^{14}C$ ages on recent lake shore plants (<4552 radiocarbon years) and animals (>2000 radiocarbon years) cited above[20]. The most proximal Baitoushan-Tianchi wiggle match ages imply substantially less (0.5%) contamination, but that tree grew, in contrast to the Taupo trees, on a well-drained mountainside, at the tree line of an open forest[47,48].

The probability distribution for the boundary between the Hukanui Pool pre- and post-ignimbrite ages had a broad peak (between 295 and 390 CE), with a long 'later' tail, resulting from the greater number of pre-ignimbrite dates. A date of 448 ± 79 CE (Fig. 4) based on the two $^{14}C$ ages in closest stratigraphic relationship with the ignimbrite, NZA7532 (1715 ± 66 BP, $\delta^{13}C$ = −27.9, on a carbonised leaf within the ignimbrite) and NZA7183 (1570 ± 68 BP, $\delta^{13}C$ = −19.3) on the tibiotarsus of a

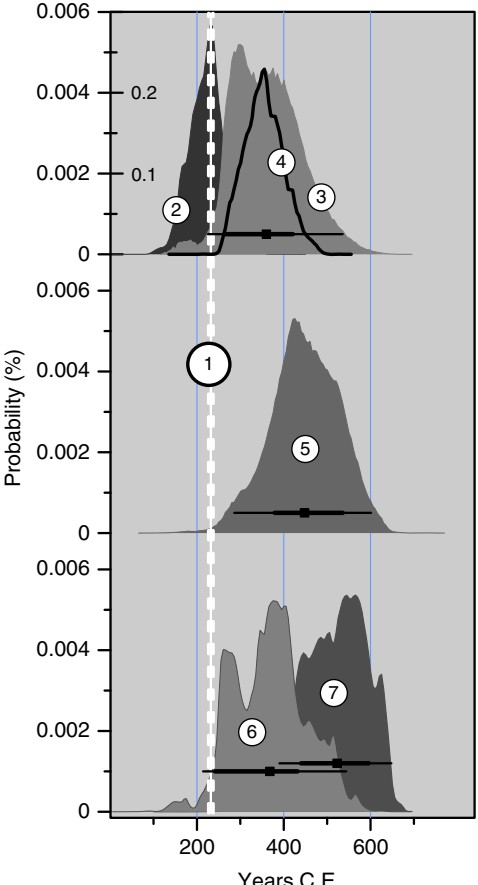

**Fig. 4** Dates for the Taupo eruption from a stratigraphic sequence outside the influence of Taupo Volcanic Zone groundwater. All dates place the Taupo eruption significantly later than 232 CE. **1**, present date for Taupo (broken line thickness, 1 s.d.); **2**, probability distribution for 25 wiggle match calibrated radiocarbon ages, combined as described in Methods; **3**, probability distribution for boundary between 8 pre- and 2 post-Taupo ages on samples from stratigraphic sequence across the Taupo ignimbrite at Hukanui, inland Hawkes Bay, North Island, New Zealand[55]; symbol and horizontal lines, mean, thick line 1$\sigma$, thin line 2$\sigma$; **4**, probability distribution for the second wiggle match ages with 5% old carbon removed; **5**, boundary between ages on carbonized leaf within the Taupo ignimbrite and *Gallirallus australis* tibiotarsus on surface of ignimbrite (symbol and horizontal lines, mean, 1 s.d., 2 s.d.). **6**, **7** probability distributions for leaf and tibiotarsus, respectively (symbol and horizontal lines, mean, thick line 1 s.d., thin line 2 s.d.)

flightless rail (Rallidae: *Gallirallus australis*) on the surface of the ignimbrite)[54] is therefore possible.

However, modelled (Oxcal "Phase", "Date" query) calibrated age ranges for extra-60 km samples were 249–337 CE (68.2%) and 206–388 CE (95.4%). The wiggle match age of 232 ± 4 CE is only just within the 95.4% calibrated date distribution, and far outside the 68.2% distribution and from the middle of both distributions. Applying the OxCal "Combination" option for the same ages on samples >60 km from the vent yielded a 68.2% calibrated age distribution of 248–339 CE (68.2%) and a shorter 95.4% distribution (245–357 CE), both excluding the present wiggle match date. The 68.2% distribution was composed of two distributions: 248–258 CE (9.7%) and 281–339 CE (58.5%). The mean combined date was 299 ± 31 CE, with a median date of 300 CE. It is possible, therefore that the offset resulting from geologic

$CO_2$ contamination may be 50–100 years (as above) up to the c. 200 years implied by the Hukanui ages. An unrecognised offset of 50–100 years in a radiocarbon-determined date for an eruption would invalidate any correlation between that eruption date and ice core records or climatic events.

**Conclusions**. Our demonstration that the [14]C samples used to date the Taupo eruption were contaminated by on-edifice magmatic carbon implies that [14]C ages on vent-proximal material for other eruptions may be similarly affected by contamination. Stratigraphic chronologies which use the Taupo and other tephras or sulphate signals in the calibration process will need to be reconsidered. The Tierra Blanca Joven eruption of Ilopango volcano, El Salvador, implicated in global climatic events[56], is one such, as well as being important in the local archaeology chronology. Further, archaeological radiocarbon chronologies in volcanically active regions, such as the Mediterranean and Central and South America, may require reanalysis. The magnitude of magmatic carbon contamination will depend on the structure of the volcanic edifice and the local climate and exposure of the dated vegetation. For example, the Santorini wiggle match ages[57] may be only marginally biased, because the wiggle match olive tree was exposed to strong winds on a mountainside in a dry environment. Errors of a few decades can be significant in some archaeological contexts, as with the 14th century Kaharoa eruption of Mt Tarawera in the Taupo volcanic zone. As a key event in Pacific prehistory chronology[6,7], an error of 50 years would significantly affect interpretation of the date of Polynesian settlement of New Zealand.

Our reanalysis of the radiocarbon ages for the Taupo ignimbrite eruption provides the first demonstration of regional biasing of [14]C eruption ages by magmatic carbon[58]. We suggest that the underlying mechanism is injection of magmatic $CO_2$ into the regional groundwater, which has a significant magmatic component[24]. The results emphasise problems associated with proximal carbon dating and highlight the importance of distal dating of eruptions[59], especially those used for stratigraphic correlation. Our methodology for identifying magma carbon bias could result in the re-dating and reinterpretation of many eruptions, with important implications for correlations with climate change, human and animal migration, and cultural adaptation. Finally, we use evidence from three caldera-forming eruptions to suggest that trends in the long-term seasonally-integrated annual records of the carbon isotopic ($\Delta^{14}C$, $\delta^{13}C$) composition of groundwater $CO_2$ in ring sequences from trees growing as part of dense forest canopies in sheltered valleys can provide evidence of injection of magma and hence inform the probability of future eruptions.

## Methods

**Data**. All radiocarbon date data are presented in Supplementary Table 1.

**Isotopic analyses**. Isotopic measurements on the analysis of the Taupo Bypass moa were performed by Isolytix Ltd, Dunedin, New Zealand (Supplementary Table 2) and tree wood were performed by Iso-Trace Ltd, Dunedin, New Zealand, with results reported in relation to standards as set out in Supplementary Table 3.

**Tree sampling—carbon stable isotopes in non-volcanic contexts**. Two 10-mm diameter cores c. 250 mm long were taken by increment borer from each of two living matai (*Prumnopitys taxifolia*) trees growing 20 m apart at 'Glenore', inland Hawkes Bay, in the eastern North Island, (176.54° E 39.254° S, altitude 736 m). Core holes were sloped c. 5° upwards to prevent retention of rain water. Both trees were sampled to only a fraction of their radius, with ring counts dating to the middle of the 18th century CE (GM1) and the late 17th century (GM2).

A prism was cut from the centre to the bark of the sectioned butt of a wind-thrown rimu (*Dacrydium cupressinum*) tree on the south bank of the Fox River,

Paparoa National Park, on the West Coast, South Island, New Zealand (171.442° E 42.044° S, altitude 68 m).

After drying at 60 °C, the cores and prism were sanded lightly along one face using progressively finer sand paper (to 1200 grit), and the ring patterns imaged using a flatbed scanner. The resulting images were enlarged to facilitate ring counts. No attempt was made to match ring width sequences between trees, as the objective was not dendrochronology per se. The time periods represented by the rings of each tree are only approximate, because we ignored potential errors such as multiple growth rings per year, years lacking rings, and counting errors associated with closely-spaced rings.

Decadal samples were cut from the cores and prism at varying intervals (c. 50 years for the prism samples), powdered in a ball mill and submitted to commercial laboratories for carbon stable isotope analysis. Results, with standards, are given in Supplementary Tables 2 & 3.

**Analysis of wiggle match date series**. Conventional wiggle match dating involves matching dates from samples separated by independently attested numbers of years (for trees as recorded by annual tree rings) to variations (wiggles) in calibration curves. If a statistically significant match is obtained, age bias is taken to be excluded[57]. However, if one or more of the ages is measured on material produced by photosynthetic uptake of $CO_2$ containing old carbon, those ages could be offset to the extent that the series could match another section of the curve entirely, unrelated to the actual life span of the tree.

The problem can be avoided, and potential circularity removed, by noting that wiggle match [14]C samples are related, to the degrees of certainty inherent in the ring counts[57], to each other by a known number of calendar years. The probability distributions of each calibrated age (in calendar time) are therefore independent measures of the calendar age of any other sample from the same tree. The date of interest for an eruption—that of the death of the tree killed by it—is the date of the last (outer) ring: this can be taken as the peak of the summed probability distributions for all the wiggle match ages, each adjusted in time by the ring count between the sample and that of the outer ring (Supplementary Fig. 6). Any level of contamination by infinite-age carbon can then be factored in by adjusting the $\Delta^{14}C$ value of the sample(s) by an appropriate amount before calibration. For the second Taupo wiggle match series of 25 [14]C ages, the date of peak probability for the published ages was identical (232 CE, Fig. 4) to the conventional wiggle match date[11], with an associated probability distribution for the eruption date (Fig. 4).

## Data availability

Data are available on request from the corresponding author.

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

## Acknowledgements

Isotopic measurements on trees outside the Taupo volcanic zone were obtained under various contracts to Palaecol Research Ltd from the New Zealand Foundation for Research, Science & Technology. B.K. acknowledges support from the New Zealand Ministry of Business Innovation and Employment Endeavour Fund ECLIPSE (Eruption or Catastrophe: Learning to Implement Preparedness for future Supervolcano Eruptions) project. We thank Deborah and Peter Turner ('Glenore') and the New Zealand Department of Conservation for permission to sample trees in Hawkes Bay and Paparoa National Park, respectively. Olivia Hyatt processed the Fox River tree section. Gayle Leaf and Doug Wall, *hapu* monitors representing Tauhara, Rauhoto Ngati Tahu, and Ngati Whaoa *iwi* approved the examination and sampling of the Taupo Bypass moa material, and Taupo District Council arranged the funding for the aDNA, radiocarbon, and stable isotopic analyses. We thank Mike Bunce (Curtin University, Perth, Western Australia) and Morten Allentoft (Natural History Museum, University of Copenhagen) for the aDNA analyses of the Taupo Bypass moa. David Hawke (Ara Institute of Canterbury) provided guidance on the interpretation of carbon stable isotope fractionation.

## Author contributions

R.N.H. identified, characterised, and interpreted the geographic and temporal trends in [14]C ages and carbon isotope ratios, provided comparative data and the radiocarbon ages for the new date for the eruption, drafted the figures (except Fig. 1b), and drafted the original MS. B.D. drafted Fig. 1b, provided information on other eruptions, and contributed to the interpretation. B.K. identified the role of groundwater, provided

volcanological context, and contributed to the interpretation. All authors contributed to the development of the manuscript and figures.

## Additional information

**Competing interests:** The authors declare no competing interests.

