## [Peer Review File · Nature Communications]

Reviewer #2 (Remarks to the Author):

The topic of the paper is important and of interest to several discussions/debates/fields as stated. The issue of effects on 14C dating from such sources of potentially old CO₂ as volcanoes has been much worried about for half a century.

It is difficult to assess a number of the statements made from the information actually provided without considerable extra work/research (not possible given timeframe allowed for review or time available to a reviewer). E.g. where are the data used for Figure 1? Ref. 14 which is cited as to imply it has, does not have these, nor Ref. 6. Assume have to go compiling from several citations.

Figure 1A seems as much to identify a known 'wobble' in the SH calibration curve in the 3rd C AD as any distance relationship. Median ages for non-monotonic 14C calibrated probability distributions are a poor guide to start (indeed the authors refer to the bimodal distributions - due to the wobble in the calibration curve - and see their Fig 1C), but nonetheless a jump/change in trajectory around 240-250 AD is to be expected just from the shape of the calibration curve with no necessary special volcanic CO₂ thesis (distance may therefore be an uncorrelated or at least not major variable - needs investigating before claiming). The comment lines 63ff on the median age of the last rings of the wobble-match - a wobble-match that fits the calibration curve taphonomy fairly well and consistently over a 200+ year period which also runs against the volcanic hypothesis since unlikely this would have been a minor time-constant effect - likewise appears to reflect a non-appreciation of the naturally changing atmospheric 14C ages as reflected in the shape of the calibration curve. Thus I see little real 'evidence' here. The authors also do not allow for the material actually dated in each case and issues of in-built age, etc. If, for example, the current data of about 232 AD were correct, and we simulate 50 radiocarbon dates for 232 AD allowing for a realistic +/-25 14C year error then we get a plot with data range not unlike the authors Fig 1A! No significant volcanic CO₂ needed! I attach a plot.

The issue of whether volcanic source CO₂ can influence 14C dates is of course important and of interest in several contexts. There have been plenty of studies showing marked (usually very large) offsets for plants growing usually very close to sources. The proposed groundwater model is rather under-explained and certainly untested (especially relevant since largely not the accepted mechanism). Most claims to date have involved an atmospheric source of entry into non-aquatic plants. The very large area effect proposed is also entirely unprecedented or demonstrated. The (good) studies cited (like ref.31) have effects usually of effectively 0 by distances of 1km, and not from groundwater.

The d13C argument is problematic. The scale of the effect proposed/required is really rather small and thus should be more or less invisible in terms of d13C (contrast where massive volcanic CO2 aging evident). Further the other studies do not support a clear relationship - detectable anyway - of d13C versus excess 14C age due to minor volcanic CO2. To use the ref. 31 study as the example (Table 2 of ref. 31), where the same plant type is compared with volcanic aging/percent volcanic carbon of 0 versus those with some effect we see that for some plant types there is no obvious pattern/response in d13C values AND where values go up (as the Taupo authors want) then really only evident where very large (many centuries and typically THOUSANDS of years of 14C extra age). This is very obviously NOT occurring in the Taupo case (where the authors end up wondering about something like a 5% effect). For example:

EFFECT % 0 6.8% 8.8% 10% 14.5% 30.4% 41.4%
 incense d13C -30.6 -29.2 -32.3 -27.1 -29.5 -30.6 -29.3

EFFECT % 0 2.2% 3.7% 11.8% 27.9% 29.0%
 fern d13C -27.8 -28.8 -30.1 -28.5 -26.1 -24.9
 +AGE +2706 +2595

EFFECT % 0 0.3% 1.5% 3.5% 12.0% 12.3% 17.5%
 Azalea -30.0 -31.4 -30.1 -27.4 -28.9 -27 -27.2
 +AGE +298 +1054 +1087 +1595

EFFECT % 0 23.6% 27.3% 35.0% 39.3%
 Heather -30.9 -27.5 -28.0 -29.8 -30.8
 +AGE +2231 +2642 +3565 +4135

Thus I find the case made undemonstrated.

In Figure 2 the authors claim to see something significant comparing d13C values from DIFFERENT species. The wiggle-match tree was *Phyllocladus trichomanoides* and thus what matai or rimu trees are doing is pretty irrelevant (and the difference in the rimu v matai only highlights the differences between species and settings - note also no consideration of likely juvenile effect in some of these samples)! On the other volcanoes discussed (Figure 3) the authors have to explain away the

Baitoushan case and also fail to note this is now very securely dated in the mid-10th century AD (<https://www.cambridge.org/core/journals/radiocarbon/article/verification-of-the-annual-dating-of-the-10th-century-baitoushan-volcano-eruption-based-on-an-ad-774775-radiocarbon-spike/72CE2CE847E3A01C0858B82487039C7D>)

The $\delta^{13}C$ of the faunal samples seem irrelevant without serious knowledge of diet and range.

Overall I see no evidence to demonstrate the claim for a significant volcanic effect on the Taupo case. The authors need to make a much stronger and explained case. Of course the topic has wide relevance if substantiated. It is almost indicative of the problem that the authors try to downplay perhaps the best known debated case: Santorini. Here suggestions of a possible volcanic CO₂ effect have been discussed for over 40 years in the literature. And so far not substantiated. Hence the authors argue maybe it was minimized here due to the windy conditions (whereas a large area - more than 11,000km² within a 60km radius - can all be supposedly consistently affected in the Taupo case despite much geographic variation at local level). But in this case distal dating has been carried out and keeps finding about the same age as most of the organics from Santorini itself (most recently: <https://www.cambridge.org/core/journals/radiocarbon/article/minoan-santorini-eruption-and-its-14c-position-in-archaeological-stratigraphic-preliminary-comparison-between-ashkelon-and-tell-eldabca/50A7C600E5FABC33D03637AB6C2B9D28>).

The proposal to re-date Taupo therefore, abandoning all the current data, on the basis of just two much later and rather non-consonant ¹⁴C dates with large measurement errors seems uncalled for on the basis of data to hand and the arguments offered. And, even then, the only really relevant date of these two seems to be NZA7532 - the leaf in the ignimbrite - whereas the other sample is from an unknown later time - and this date includes the wiggle-match date within its 95.4% hpd range of 223-538 Cal AD and is close to catching the same wiggle in the SH ¹⁴C calibration curve in the 3rd century AD.

A much stronger and better justified case would be needed, linking ideas/hypotheses to data observed and mechanisms known/investigated. I do not think this paper as presented should be published.

Reviewer #3 (Remarks to the Author):

This manuscript reports a thorough investigation of radiocarbon dating the Taupo First Millennium eruption in North Island, New Zealand, in relation to the intricate complications of magmatic carbon additions. It is an important and novel contribution, very much worth publishing. All figures are clear and necessary in relation to the text.

As the manuscript is submitted to an interdisciplinary international journal with a wide global readership, I strongly recommend the authors to write a concise paragraph, to be inserted on page 2 after the opening paragraph (line 49), giving a brief overview of the Taupo Volcano. This overview should include a description of the type of volcano in volcanological terms, its large caldera, filled by a lake that is the largest lake in New Zealand. Also include the type of climate, average annual rainfall amount and the groundwater regime in the area.

The scientific aspects of the investigation are comprehensive, including many measurement results of tree samples covered by volcanic products of this eruption up to about 180 km away from the vent.

To further strengthen their case, I recommend the authors to include one or two sentences, with references, showing that all the ejected volcanic material covering these trees along this range of 180 km are undoubtedly from the same eruption.

Understandably, the authors mention several times that their significant findings may also require reevaluation of radiocarbon dating results of other volcanic eruptions.

Page 7 of their manuscript, lines 213-215: "Our methodology for identifying magma carbon bias could result in the re-dating and reinterpretation of many eruptions, with important implications for correlations with climate change, human and animal migration, and cultural adaptation."

However, the word "many" may be an overstatement, because there are various different types of volcanoes in the world, having quite different productions of magmatic carbon outgassing, and also situated in different climatic and environmental settings with respect to groundwater.

The authors refer to the Minoan Santorini eruption as an example of another large eruption that has great chronological significance in relation to historical-archaeological connections in the eastern Mediterranean region. This is fine. However, the authors should include in this context the most updated references in relation to a possible age offset by magmatic CO₂. An article was published in *Antiquity*, showing that tsunami deposits in Crete, caused by the Minoan eruption and also containing volcanic ash from this eruption, have the same uncalibrated radiocarbon BP age as the

uncalibrated BP date for the eruption based on short-lived plant material from the archaeological site of Akrotiri on Thera, close to the vent. This result suggests that at Santorini, there is no measurable magmatic CO₂ offset in the vegetation. Hence the radiocarbon dates for the Santorini eruption do not suffer from this effect. The reference is: Bruins, H.J. and van der Plicht, J., 2014. The Thera olive branch, Akrotiri (Thera) and Palaikastro (Crete): comparing radiocarbon results of the Santorini eruption. *Antiquity* 88: 282–287.

Indeed, a recent study of modern vegetation at Santorini showed that there is no such offset in the great majority of measurements (with one exception). The reference is:

Fernandes, R., Dreves, A., Klontza-Jaklova, V., Cook, G., 2016. Investigating potential radiocarbon offsets in Mediterranean plants: Implications for the dating of the Thera eruption. Poster, 8th International Symposium, 14C & Archaeology, 27 June–1 July, 2016, Edinburgh, Scotland, UK.

For your convenience, both publications are attached.

In conclusion, this is a very good manuscript with very significant results. The text requires, in my opinion, only a few amendments, as indicated above.

Our detailed responses to the points raised by the reviewers are set out below.

REVIEWER 2

Comment

Figure 1A seems as much to identify a known 'wiggle' in the SH calibration curve in the 3rd C AD as any distance relationship. Median ages for non-monotonic ^{14}C calibrated probability distributions are a poor guide to start (indeed the authors refer to the bimodal distributions - due to the wiggle in the calibration curve - and see their Fig 1C), but nonetheless a jump/change in trajectory around 240-250 AD is to be expected just from the shape of the calibration curve with no necessary special volcanic CO_2 thesis (distance may therefore be an uncorrelated or at least not major variable - needs investigating before claiming).

Response

The reviewer raises an important potential explanation for the age-distance relationship. However, our more detailed representation of the data shows that the suggested alternative cannot explain the striking relationship. We now show the probability distributions on Figure 1A, showing that most of the calibrated age distributions on samples beyond 60 km from the vent are monotonic, with their peak probabilities (and most of the distributions) after the wiggle match age. Two dates with "old" median ages on material from > 60 km from the vent are: (1) from material from close to the Waikato River and hence within the area affected by geologic CO_2 (Beavan-Athfield *et al.* 2001); and (2), which has a bimodal probability distribution, resulting from the "3rd century wiggle" which includes a substantial area of probability within the range of the monotonic distributions of the younger, more distant, age measurements.

Figure 1B, by showing clearly that the ages are geographically arrayed by azimuth and not just by distance, demonstrates that the wiggle in the calibration curve cannot be responsible for the variation in ages. The atmospheric CO_2 wiggle cannot vary geographically over such short distances.

Figure 1C has been modified to emphasise the abrupt change in the area of highest probability at c. 60 km from the vent. The summed distributions are still diatonic but all groups > 60 km have much greater probabilities in the 4th century CE. The new plot emphasises the major shift between the summed probabilities for samples < 60km from the vent, and the summed probabilities for materials > 60 km from the vent.

Overall, the reviewer's claim that distance is "uncorrelated or not major variable" is surprising in the face of the evidence presented in Fig. 1B. We reinforce this point here with Response Figure 1 below, which compares interpolated age with rift geometry and topography. We therefore reject the reviewer's geographic comment, pointing out that all the youngest ages are outside the rift.

Response Figure 1

Comment

The comment lines 63ff on the median age of the last rings of the wiggle-match - a wiggle-match that fits the calibration curve taphonomy fairly well and consistently over a 200+ year period which also runs against the volcanic hypothesis since unlikely this would have been a minor time-constant effect - likewise appears to reflect a non-appreciation of the naturally changing atmospheric ¹⁴C ages as reflected in the shape of the calibration curve. Thus I see little real 'evidence' here.

Response

We additionally show that the wiggle match matches our section of the curve equally well (as shown in Response Figure 2 below), perhaps better. Of course, the naturally changing atmospheric ¹⁴C is quasi-cyclical which exacerbates this kind of problem with wiggle match age series.

Response Figure 2

Comment

where are the data used for Figure 1? Ref. 14 which is cited as to imply it has, does not have these, nor Ref. 6. Assume have to go compiling from several citations.

Response

All pre-wiggle match ^{14}C ages are cited in reference 14 and others in References 42 and 43. For convenience, a fully referenced database of ages is now provided as Supplementary Information Table 1.

Comment

The authors also do not allow for the material actually dated in each case and issues of in-built age, etc. If, for example, the current data of about 232 AD were correct, and we simulate 50 radiocarbon dates for 232 AD allowing for a realistic ± 25 ^{14}C year error then we get a plot with data range not unlike the authors Fig 1A! No significant volcanic CO_2 needed! I attach a plot.

Response

If the material dated is considered (see Response Figure 3), the wood and charcoal datasets that contain sufficient samples for distance relationships to be evaluated both display the same relationship as the aggregate dataset. The comment regarding material variations must also therefore be rejected. We do not show this on Fig. 1 to avoid excessive complexity of that figure. However, it is now clearly shown for the purposes of readers by using a colour ramp in Supplementary Information Table 1 to demonstrate the correlation of distance and age.

We also refute the reviewer's plot, which we show does not resemble the data range of Fig. 1A, producing only about half the spread of ages, and including none of the younger ages, as is clearly seen in Response Figure 3. This figure is now included in the Supplementary Information.

Response Figure 3

Comment

The issue of whether volcanic source CO_2 can influence ^{14}C dates is of course important and of interest in several contexts. There have been plenty of studies showing marked (usually very large) offsets for plants growing usually very close to sources. The proposed groundwater model is rather under-explained and certainly untested (especially relevant since largely not the accepted mechanism). Most claims to date have involved an atmospheric source of entry into non-aquatic plants. The very large area effect proposed is also entirely unprecedented or demonstrated. The (good) studies cited (like ref.31) have effects usually of effectively 0 by distances of 1km, and not from groundwater.

Response

The reviewer takes issue with the groundwater model we invoke. In fact, in recognition of the importance of differential groundwater CO_2 concentrations for tree growth is not new: Amiro and Ewing (1992) measured plant uptake of ^{14}C via the roots and found that it was independent of the photosynthetic rate and primarily controlled by transpiration rate and nutrient solution (groundwater) concentration. Saurer et al. (2003) traced the uptake of ^{14}C from CO_2 springs into tree rings. Although the method of incorporation is not clear, a mix of atmospheric and groundwater uptake was suggested by the continued influence some 150 m from the spring, despite the open habitat. Vejpusťková et al. (2016) showed that high groundwater CO_2 concentrations enhance wood growth over lower CO_2 concentration sites, even within a single stand of trees with steep CO_2 concentration gradients and largely homogenized canopy CO_2 concentrations. Note that this was in a situation with adjacent moffettes (ground vents from which CO_2 and other gases escape). The results led those authors to conclude that

“trees can be fertilized not only by elevated atmospheric CO_2 but also when fed with CO_2 via the roots”.

These published data are consistent with our interpretation. Regarding the scale of the affected area, few studies have addressed this problem in an analogous, low relief caldera setting. However, a study in the Long Valley Caldera of California documented anomalous ^{14}C ages in plants 65 km downstream of the caldera (Reid *et al.* 1998). We now cite this study in the manuscript, providing another key reference in support of our groundwater model, and

providing the additional explanation and evidence of testing that the reviewer suggests is missing.

As the reviewer notes, the area involved could be considered large, but an extensive body of groundwater with high concentrations of geologic-sourced DIC is consistent with data from groundwater in the TVZ (Giggenbach 1995). Continuity of the groundwater through the TVZ is assured by its geology, as now set out in the paragraph added to the text: “The TVZ hosts a 4.2 GW geothermal system that is fed from shallow (2-7 km deep) heat sources (Wilson & Rowland 2016). The region receives about 1200 mm annual rainfall, which recharges aquifers developed in variably permeable pyroclastic deposits that are increasingly fractured with age/depth and interspersed with lacustrine or paleosol aquitards (Hadfield *et al.* 2001).” As to whether the area is of unprecedented size, this is true compared to the more commonly studied stratocone volcanoes. However, calderas in general, and the Taupo volcanic zone specifically, support evidence for unprecedented amounts of subsurface magma and magmatic degassing, producing an unprecedented area of carbon bias.

Comment

The $\delta^{13}\text{C}$ argument is problematic. The scale of the effect proposed/required is really rather small and thus should be more or less invisible in terms of $\delta^{13}\text{C}$ (contrast where massive volcanic CO_2 aging evident). Further the other studies do not support a clear relationship - detectable anyway - of $\delta^{13}\text{C}$ versus excess ^{14}C age due to minor volcanic CO_2 . To use the ref. 31 study as the example (Table 2 of ref. 31), where the same plant type is compared with volcanic aging/percent volcanic carbon of 0 versus those with some effect we see that for some plant types there is no obvious pattern/response in $\delta^{13}\text{C}$ values AND where values go up (as the Taupo authors want) then really only evident where very large (many centuries and typically THOUSANDS of years of ^{14}C extra age). This is very obviously NOT occurring in the Taupo case (where the authors end up wondering about something like a 5% effect). For example:

EFFECT %	0	6.8%	8.8%	10%	14.5%	30.4%	41.4%
incense $\delta^{13}\text{C}$	-30.6	-29.2	-32.3	-27.1	-29.5	-30.6	-29.3
EFFECT %	0	2.2%	3.7%	11.8%	27.9%	29.0%	
fern $\delta^{13}\text{C}$	-27.8	-28.8	-30.1	-28.5	-26.1	-24.9	
+AGE	+2706	+2595					
EFFECT %	0	0.3%	1.5%	3.5%	12.0%	12.3%	17.5%
Azalea $\delta^{13}\text{C}$	-30.0	-31.4	-30.1	-27.4	-28.9	-27	-27.2
+AGE	+298	+1054	+1087	+1595			
EFFECT %	0	23.6%	27.3%	35.0%	39.3%		
Heather $\delta^{13}\text{C}$	-30.9	-27.5	-28.0	-29.8	-30.8		
+AGE	+2231	+2642	+3565	+4135			

Thus I find the case made undemonstrated

Response

There are two different uses to which $\delta^{13}\text{C}$ measurements are put: first, and this is the point of the reviewer's comment, is in the generation of the ^{14}C age itself. If we were proposing that the $\delta^{13}\text{C}$ values were driving a bias in radiocarbon ages this would indeed be problematic, to say the least. However, we are *not* proposing that at all: we invoke the second, and **independent**, use of $\delta^{13}\text{C}$ measurements, which is to determine the **source** of the ^{14}C -free carbon contamination in the carbon whose radiometric age is being measured. We indeed clearly

propose that the bias in ages is caused by an up to 5% contamination of the measured carbon by ^{14}C -free carbon, and not by a 5% (or any other) shift in the $\delta^{13}\text{C}$ of the carbon.

To reiterate, **the only connection we make between $\delta^{13}\text{C}$ values and the ^{14}C ages is that the $\delta^{13}\text{C}$ values can indicate the source of the contamination of the dated carbon. We do not attribute the offset in ^{14}C ages to the effect of the $\delta^{13}\text{C}$ values on the dating process itself.** We have clarified this in the text.

Comment

In Figure 2 the authors claim to see something significant comparing $\delta^{13}\text{C}$ values from DIFFERENT species. The wiggle-match tree was *Phyllocladus trichomanoides* and thus what matai or rimu trees are doing is pretty irrelevant (and the difference in the rimu v matai only highlights the differences between species and settings - note also no consideration of likely juvenile effect in some of these samples)!

Response

The data on matai and rimu trees are not irrelevant, as we show below. In addition, the “juvenile effect” is clearly shown by the shift in $\delta^{13}\text{C}$ values for the rimu tree in Figure 2, and in Response Figure 4. In contrast, the wiggle match tree displays no such trend from juvenile to canopy adult, despite the measurements including rings from its earliest (subcanopy) growth.

In the MS we present three lines of evidence in support of our contention that the $\delta^{13}\text{C}$ values in the wiggle match wood cellulose are aberrant, indicating a contribution of geologic CO_2 in the TVZ ground water. These are: the actual $\delta^{13}\text{C}$ values of leaves and wood of a range of tree taxa, including a species of *Phyllocladus*, in comparison with those of the wiggle match tree cellulose; the trend in $\delta^{13}\text{C}$ values across tree stems related to the change from sub-canopy saplings to canopy adults; and the $\delta^{13}\text{C}$ values for moa feeding in pre-human vegetation.

The three main potential drivers of modern foliage $\delta^{13}\text{C}$ values versus old wood, in similarly wet environments are: the Suess Effect (which is taken into account in Response Figure 4, where all values are reduced to pre-1850 CE values); the foliage-wood fractionation ($\leq 2\text{‰}$); and inter-taxon differences (up to c. 3 ‰). Even summed at their maximum values, the combined differences will not bridge the gap between the leaf values shown in Response Figure 4 (which include a species of *Phyllocladus*) and the wiggle match tree wood cellulose. In particular, inter-taxic differences would not bridge the 8‰ difference between the early wood of the rimu and the wiggle match tree.

The juvenile effect for plants growing beneath a canopy is for the leaves and wood to exhibit $\delta^{13}\text{C}$ values lower than those of tissues laid down when the plant reaches the canopy (Response Figure 4) or the canopy is removed (matai trees in Figure 2). The effect is general, in tropical and temperate ecosystems, e.g., (Medina & Minchin 1980; van der Merwe & Medina 1991; Brooks *et al.* 1997; Cerling *et al.* 2004). So, instead of being within 2‰ of the final rings, they should be the lowest of all $\delta^{13}\text{C}$ values by 4-5 ‰ at least within a single tree growing, as the *Phyllocladus trichomanoides* tree is known to have been, in a dense podocarp forest in a valley (Clarkson *et al.* 1988; Clarkson *et al.* 1992). The record should exhibit the same “hockey stick” curvature as those for New Zealand forests in Response Figure 4.

The reviewer raises the question of inter-taxic differences in $\delta^{13}\text{C}$ value. These exist, as is generally recognised, but are minor (maximum $\sim 3\text{‰}$). It is also generally recognised that the effects of light and carbon source have much greater effects, with one of the best known being the so-called “canopy effect” (Heaton 1999; Badeck *et al.* 2005; Gessler *et al.* 2014). Plants

growing beneath a closed canopy (even of dense grass) display much lower $\delta^{13}\text{C}$ values than those with full access to atmospheric equilibrium CO_2 . In Response Figure 4, it can be seen that the inter-taxic effect is of the same order as that within each taxon for foliar values from 0.5 to 3.5 m above ground.

The wiggle match tree $\delta^{13}\text{C}$ values fall well outside those for New Zealand trees, including another species of *Phyllocladus*, whether under the canopy and juvenile or within the canopy or removed from canopy influences (Response Figure 4). All the gymnosperms sampled, including both rimu (*Dacrydium*) and *Phyllocladus alpinus*, have similar (low/depleted) $\delta^{13}\text{C}$ values when growing beneath closed canopies. This is one of the key factors for interpretation of carbon stable isotopic ratios in environmental isotopic research (Ambrose & DeNiro 1986).

Response Figure 4

The sub-canopy data presented as points and means are for leaves which are up to 4‰ below that for wood in an adult tree (Heaton 1999). For juveniles beneath a canopy, the leaf-wood differential in $\delta^{13}\text{C}$ is less.

Source CO_2 has much greater affect than any inter-taxic effect. Beneath a canopy, plant foliage typically exhibits $\delta^{13}\text{C}$ values between -27 and -33‰, far (8-10‰) distant from the -22‰ of the early growth of the wiggle match tree wood cellulose, even allowing for the c.2‰ foliage/wood fractionation.

Even growing outside a canopy – and for trees within forest, after they have reached the canopy – the wood $\delta^{13}\text{C}$ values peak at c -23‰, not 20‰ as in the wiggle match tree (Figure 2). The extremely high (-22 to -20‰) values for the wood cellulose of the wiggle match tree are accompanied by a lack of an increasing trend with the tree's growth over many decades from an entirely sub-canopy sapling to an adult tree with its foliage arrayed within the dense canopy. The high values and lack of a trend indicate a considerable contribution of carbon with an even higher $\delta^{13}\text{C}$ value, provided by the c. -9‰ of the DIC. In referring to percentages of amount of

“dead” carbon necessary to offset the ages, the addition of < 5% of ^{14}C -free carbon outgassed from the groundwater is sufficient to shift the age by several centuries in the time frame of 1500-2000 years BP. The level of contamination by ^{14}C -free carbon in the groundwater is confirmed by the 2000-year-plus date offsets in the duck and rat from the Lake Taupo food web.

In summary, the rimu and matai tree records are not irrelevant. They show the normal values and trends in $\delta^{13}\text{C}$ values of trees within a New Zealand forest. Their narrow range and extreme elevation, especially when the tree was young and growing beneath the canopy, are aberrant and cannot be explained by inter-taxic differences as suggested by Reviewer 2. Indeed, the inter-taxic differences in $\delta^{13}\text{C}$ implied by Reviewer 2 would negate all environmental stable isotopic analyses. These issues are now dealt with in an additional paragraph and referencing in the text.

Comment

On the other volcanoes discussed (Figure 3) the authors have to explain away the Baitoushan case and also fail to note this is now very securely dated in the mid-10th century AD (<https://www.cambridge.org/core/journals/radiocarbon/article/verification-of-the-annual-dating-of-the-10th-century-baitoushan-volcano-eruption-based-on-an-ad-774775-radiocarbon-spike/72CE2CE847E3A01C0858B82487039C7D>)

Response

The paper we failed to note was published after we submitted our paper. It is interesting, however, that this new paper completely supports Xu *et al.*'s age (Xu *et al.* 2013), which is collected from 24 km distance and clearly offset from the calibration curve, as noted by the authors *who propose magmatic CO_2 as a possible cause*. The offset is only obvious because of the favourable shape of the calibration curve, as we note in our manuscript.

Response Figure 5

On the point of favourable shapes, Rabaul yielded a remarkably poor wiggle match (Response Figure 5). In part this may be because (i) it consists of only 5 radiocarbon ages; (ii) the choice of calibration curve is ambiguous; (iii) one of these 4 ages is considered somewhat of an outlier by the paper's authors and greatly affects the fit, but not the age. But is it an outlier?

It is interesting that the fit is very good if (a) the radiocarbon ages are too old by 135 years, and (b) the equatorial calibration curve is a mix of the northern and southern hemisphere curves, with lows equivalent to IntCal13 and high peaks equivalent to SHCal13 (Response Figure 5). Given that the general elevation of the SHCal13 arises from the strength of the degassing in the Southern Ocean, it does not seem unreasonable that a variable influence may be observed at a very low latitude southern hemisphere site that is influenced by the ITCZ.

Perhaps it is telling that two of four wiggle matched volcanoes, including multiple wiggle matches from Baitoushan-Tianchi, (i.e. actually amounting to >50% of individual published volcanic wiggle matches, and 66% if the dubious age at Rabaul is included) have shown demonstrable offsets. If these wiggle matches display offsets, why not Taupo, for which we also present a suite of additional compelling data with a causative and consistent explanation?

As for “explaining away”, we reiterate that our topographic interpretation is consistent across examples and supported by: (1) the evidence of water-borne CO₂ contaminating ages 65 km distant from the Long-Valley Caldera rim (a recessed topographic feature), which is a similar situation to the Taupo region; (2) only minimal offset at Baitoushan (a variably vegetated positive topographic feature); and (3) no obvious offset at Santorini, an exposed positive topographic feature.

Comment

The $\delta^{13}\text{C}$ of the faunal samples seem irrelevant without serious knowledge of diet and range.

Response

We contend that the faunal sample $\delta^{13}\text{C}$ values are indeed relevant, as they show offsets from terrestrial or freshwater values expected in areas unaffected by volcanism. Contra Reviewer 2's comment, there is no doubt about the diets and feeding ranges of these species. New Zealand scaup (*Aythya novaeseelandiae*) feed on bottom-living invertebrates and macrophytes in lakes, streams, and coastal lagoons (Heather & Robertson 1996; Wakelin 2004). Norway rats (*Rattus norvegicus*) are opportunistic omnivores (Moors 1990). In coastal environments, as on the shores of Lake Taupo, they eat shoreline foods and those obtained by diving: they swim readily (Moors 1990).

The $\delta^{13}\text{C}$ value of the scaup bone places it close to those for recent scaup from Rotorua lakes, in a separate distribution to those of fossil and recent individuals of the same species feeding in water lacking geologic CO₂ (Response figure 6, below, from RNH, M. Williams, K. Rogers, unpublished data). The high $\delta^{13}\text{C}$ value for the Lake Taupo scaup accords with those of the Lake Taupo water DIC (which indicate a high titre of geologic CO₂ and hence a high titre of ¹⁴C-dead carbon). The $\delta^{13}\text{C}$ value of the “old” Norway rat indicates that it was part of the Lake Taupo freshwater food web (Response Figure 6).

The very “old” ¹⁴C ages for the Norway rat and New Zealand scaup resulted from the animals' obtaining much if not all of their dietary carbon from the Lake Taupo food web.

Beavan-Athfield *et al.* (2001) point out that as the Waikato River flows out of Lake Taupo, ¹⁴C ages on material downstream in this major waterway might well be affected by the “old” carbon in the water. Of course, the lake and river waters are not confined by the lake and river beds and their DIC must reflect that of the groundwater in the adjacent land mass, which

includes the area where the wiggly match tree was growing. This effect is also likely to be responsible for the “old” age on the sample from Arapuni (1 in revised Figure 1A).

The duck, rat, and wiggly match tree were all deriving a proportion of their carbon from the dissolved inorganic carbon reservoir in the same water mass, the groundwater of the TVZ.

Response Figure 6

Comment

Overall I see no evidence to demonstrate the claim for a significant volcanic effect on the Taupo case.

The authors need to make a much stronger and explained case. Of course the topic has wide relevance if substantiated. It is almost indicative of the problem that the authors try to downplay perhaps the best known debated case: Santorini. Here suggestions of a possible volcanic CO2 effect have been discussed for over 40 years in the literature. And so far not substantiated. Hence the authors argue maybe it was minimized here due to the windy conditions (whereas a large area - more than 11,000km2 within a 60km radius - can all be supposedly consistently affected in the Taupo case despite much geographic variation at local level). But in this case distal dating has been carried out and keeps finding about the same age as most of the organics from Santorini itself (most recently:

<https://www.cambridge.org/core/journals/radiocarbon/article/minoan-santorini-eruption-and-its-14c-position-in-archaeological-strata-preliminary-comparison-between-ashkelon-and-tell-eldabca/50A7C600E5FABC33D03637AB6C2B9D28>).

Response

We believe that we provided a strong case which we have clarified and explicated further in these responses.

As noted above, the authors of the most recent Baitoushan paper go so far as to suggest the presence of geologic carbon bias. The geographic variation is moot when most of the samples

come from swampy forests inside a graben. Observables so far – swampy forests in a graben = big offsets (and demonstrated groundwater carriage of old carbon in both Taupo and Long Valley); forests on a mountain side = less offset but still significant; unforested Mediterranean mountainside = minimal to no offset.

There is, therefore, no attempt to “downplay” the Santorini example, as that simply exemplifies a situation where, for the reasons canvassed above, a significant geologic CO₂ offset to the dating would *not* be expected because the wiggle match tree was almost certainly on an unforested mountainside in the eastern Mediterranean. This in no way argues against such offsets being present in different volcanic settings.

The considerable geographic extent of the effect in the TVZ is, as with the Long Valley example, a result of the nature of the TVZ geomorphology, and the extent of the groundwater system that surrounds and supports a 180 m-deep, 616 km², lake and major river systems. As we now note in the MS, the “aquifers [are] developed in variably permeable pyroclastic deposits that are increasingly fractured with age/depth and interspersed with lacustrine or paleosol aquitards”, which implies a continuity of groundwater over the area.

Comment

The proposal to re-date Taupo therefore, abandoning all the current data, on the basis of just two much later and rather non-consonant ¹⁴C dates with large measurement errors seems uncalled for on the basis of data to hand and the arguments offered. And, even then, the only really relevant date of these two seems to be NZA7532 - the leaf in the ignimbrite - whereas the other sample is from an unknown later time - and this date includes the wiggle-match date within its 95.4% hpd range of 223-538 Cal AD and is close to catching the same wiggle in the SHCal13 ¹⁴C calibration curve in the 3rd century AD.

Response

The “other sample” was a bone physically resting on Taupo ignimbrite in an enclosed rock shelter and was noted at the time of excavation as having been deposited immediately after the eruption – potentially a few decades. The sediment above the bone was typical of that in the remainder of the site stratigraphy, a consolidated granular matrix in which overhangs of > 90 degrees were easy to sustain. Deposition of cave sediment by decomposition of the roof slabs in the site averaged 0.11 mm year⁻¹ (Holdaway & Beavan 1999) so at least 5 mm of consolidated sediment would be expected to have accumulated below the bone if it had been deposited 50 years after the eruption. The lack of this pre-deposition layer suggests that the bone was deposited within a decade or two of the eruption.

REVIEWER 3

Comment

This manuscript reports a thorough investigation of radiocarbon dating the Taupo First Millennium eruption in North Island, New Zealand, in relation to the intricate complications of magmatic carbon additions. It is an important and novel contribution, very much worth publishing. All figures are clear and necessary in relation to the text.

Response

Thank you.

Comment

As the manuscript is submitted to an interdisciplinary international journal with a wide global readership, I strongly recommend the authors to write a concise paragraph, to be inserted on page 2 after the opening paragraph (line 49), giving a brief overview of the Taupo Volcano. This overview should include a description of the type of volcano in volcanological terms, its large caldera, filled by a lake that is the largest lake in New Zealand. Also include the type of climate, average annual rainfall amount and the groundwater regime in the area.

Response

We acknowledge the need to provide more background for an international readership and have inserted a paragraph as suggested, integrated with the textual revision occasioned by the change to *Nature communications* format, i.e. removal of references from a shortened abstract, and their insertion into the modified text introducing the paper.

References cited in responses

- Ambrose, S. H.; DeNiro M. J. 1986. The isotopic ecology of East African mammals. *Oecologia* 69: 395-406.
- Amiro, B. D.; Ewing L. L. 1992. Physiological conditions and uptake of inorganic carbon-14 by plant roots. *Environmental and Experimental Botany* 32: 203-211.
- Badeck, F. W.; Tcherkez G.; Nogues S.; Piel C.; Ghashghaie J. 2005. Post-photosynthetic fractionation of stable carbon isotopes between plant organs—a widespread phenomenon. *Rapid communications in mass spectrometry* 19: 1381-1391.
- Beavan-Athfield, N. R.; McFadgen B. G.; Sparks R. J. 2001. Environmental Influences on Dietary Carbon and $\delta^{13}C$ Ages in Modern Rats and Other Species. *Radiocarbon* 43: 7-14.
- Brooks, J. R.; Flanagan L. B.; Buchmann N.; Ehleringer J. R. 1997. Carbon isotope composition of boreal plants: functional grouping of life forms. *Oecologia* 110: 301-311.
- Cerling, T. E.; Hart J. A.; Hart T. B. 2004. Stable isotope ecology in the Ituri Forest. *Oecologia* 138: 5-12.
- Clarkson, B. R.; Clarkson B. D.; Patel R. N. 1992. The pre-Taupo eruption (c. AD 130) forest of the Benneydale-Pureora district, central North Island, New Zealand. *Journal of the Royal Society of New Zealand* 22: 61-76.
- Clarkson, B. R.; Patel R. N.; Clarkson B. D. 1988. Composition and structure of forest overwhelmed at Pureora, central North Island, New Zealand, during the Taupo eruption (c. AD 130). *Journal of the Royal Society of New Zealand* 18: 417-436.
- Gessler, A.; Ferrio J. P.; Hommel R.; Treydte K.; Werner R. A.; Monson R. K. 2014. Stable isotopes in tree rings: towards a mechanistic understanding of isotope fractionation and mixing processes from the leaves to the wood. *Tree physiology* 34: 796-818.
- Giggenbach, W. F. 1995. Variations in the chemical and isotopic composition of fluids discharged from the Taupo Volcanic Zone, New Zealand. *Journal of Volcanology and Geothermal Research* 68: 89-116.
- Hadfield, J.; Nicole D.; Rosen M.; Wilson C. J. L.; Morgenstern U. 2001. *Hydrogeology of Lake Taupo catchment : phase I*. Hamilton East [N.Z.]: Environment Waikato.
- Heather, B. D.; Robertson H. A. 1996. *The field guide to the birds of New Zealand*. Auckland: Viking.
- Heaton, T. H. 1999. Spatial, species, and temporal variations in the $\delta^{13}C/\delta^{12}C$ ratios of C3 plants: implications for palaeodiet studies. *Journal of Archaeological Science* 26: 637-649.
- Medina, E.; Minchin P. 1980. Stratification of $\delta^{13}C$ values of leaves in Amazonian rain forests. *Oecologia* 45: 377-378.
- Moors, P. J. (1990) Norway rat. 192-206. In: King CM (ed.) *The handbook of New Zealand mammals*. Auckland: Oxford University Press, 192-206.
- Reid, J. B.; Reynolds J. L.; Connolly N. T.; Getz S. L.; Polissar P. J.; Winship L. J.; Hainsworth L. J. 1998. Carbon isotopes in aquatic plants, Long Valley Caldera, California as records of past hydrothermal and magmatic activity. *Geophysical Research Letters* 25: 2853-2856.
- Saurer, M.; Cherubini P.; Bonani G.; Siegwolf R. 2003 Tracing carbon uptake from a natural CO_2 spring into tree rings: an isotope approach. Canada: HERON PUBLISHING, 997.
- van der Merwe, N. J.; Medina E. 1991. The canopy effect, carbon isotope ratios and foodwebs in Amazonia. *Journal of Archaeological Science* 18: 249-259.
- Vejpustková, M.; Thomalla A.; Čihák T.; Lomský B.; Pfanz H. 2016. Growth of *Populus tremula* on CO_2 -enriched soil at a natural mofette site. *Dendrobiology* 75: 3-12.
- Wakelin, M. 2004. Foods of the New Zealand dabchick (*Poliiocephalus rufopectus*) and New Zealand scaup (*Aythya novaeseelandiae*). *Notornis* 51: 242-245.
- Wilson, C. J. N.; Rowland J. V. 2016. The volcanic, magmatic and tectonic setting of the Taupo Volcanic Zone, New Zealand, reviewed from a geothermal perspective. *Geothermics* 59: 168-187.
- Xu, J.; Pan B.; Liu T.; Hajdas I.; Zhao B.; Yu H.; Liu R.; Zhao P. 2013. Climatic impact of the Millennium eruption of Changbaishan volcano in China: New insights from high-precision radiocarbon wiggle-match dating. *Geophysical Research Letters* 40: 54-59.

Reviewer #2 (Remarks to the Author):

The authors in their detailed response and revisions to their main text, figures and supporting material now go a very long way to meeting a number of the queries this reviewer had from the initial submission. They have either adequately dealt with several possible issues raised, or provided additional information which clarifies what were queries/questions before, or gives necessary information to assess.

Some issues remain and should be addressed before publication.

1. The authors propose an alternative wiggle match - suggested as showing a ~240 14C offset (SI Fig. 6). They suggest this is equally valid. But this does not seem to be the case - and the authors offer no good evidence for this view. I assume they used the Wk data set from Hogg et al. 2011? If so, then a wiggle-match adjusted with a 240 14C factor does NOT offer a very good fit versus SHCal13 and when compared to the modelling reported by Hogg et al. It fails a X2 test and the OxCal Aoverall value is only 10.2 < 60 and 36% of the sample offer poor individual agreement values (versus just 8% in the Hogg et al. model). The authors need to provide some details and backup of their claim that there is a viable alternative.

2. This leads on to a second issue. Do the authors really envisage the apparent stability (versus variability) of the groundwater effect over more than 200 years - which seems unlikely or at least in need of further demonstration - and does not seem supported by the failure of the stable adjustment wiggle-match to offer a satisfactory fit versus the calibration curve.

3. The data leading to Fig. 1 suggest that the inflection point between some potential substantial 14C offset and a relatively stable situation (and one therefore assumes insignificantly affected results) is at around 60km distance from the assumed vent. The wiggle match data came from around 50km or so from the vent towards this inflection point range - if the authors state a distance I have failed to spot it as obvious in their paper, so I take the number from Hogg et al. 2011. Thus finding a large - 240 14C year offset - this far away seems to run somewhat against their own findings (and of their data points in Fig. 1 for ~50km where two are in their OK set above the red dashed line and only one is below). The Hogg et al. work used a single tree (FS066) - one obvious test would be to try another tree or better pool wood from several trees and then redo the wiggle-match and see how consistent the data are.

4. Most of the other 14C dates employed are not recent and have quite large measurement errors (i.e. arguably all but one date in SI Table 1.) While this does not invalidate the work, it is problematic

when considering medians of 14C calibrated ranges, as these ranges will be much larger and likely multi-modal where data have large measurement errors and bias more easily enters in taking a point value as indicative. Much more concerning, I fail to see that the 'other' data in Figure 1 do not more or less support the existing date. If we model the dates from >60km distance as a Phase and ask a Date query using SHCal13, we get CalAD 249-337 (68.2%) and Cal AD 206-388 (95.4%), a range pretty well centered around the current estimate: Figure 1 attached.

5. There are no dates on short-lived samples in the <60km set! There is no good information on how much in-built age factor applies for the wood/charcoal in this set. Based on the information available from the dendro studies, the answer could easily be substantial! This could easily account for at least a part and even substantial part of the supposed 'old' offset. There are clearly some older ages in this 'closer' set - the issue is whether, given a fairly small and non-characterized sample set (samples not all stated definitely to be outermost rings, for example), it could merely be chance that there are a few cases of older wood/inner tree-rings in this closer set? To consider we could model all the dates in SI Table 1 on charcoal or wood as a Phase in OxCal using the Charcoal outlier model (Bronk Ramsey 2009) - see Figure 2 attached. The data are ordered by distance from the assumed vent (first value in label is km distance from SI Table 1). I see a pretty typical charcoal distribution. A few older to somewhat older dates and then a lot of dates largely similar and best defining a TPQ. The red dashed box shows a range from ca. AD200-350 which captures the main probability of most of the samples from both <60km and >60km. Yes, there are some older ages in the <60km set but nothing beyond inner wings/old wood as typical from 14C dating random sets of charcoal - there are some instances (up to 5) in the >60km set too. But overall they look quite 'typical' of such data from other contexts when dealing with wood or charcoal from long-lived trees. The TPQ from the set is AD303-393 (68.2%) and AD259-435 (95.4%). This could be fine with the existing date (which is also a TPQ) or it could suggest a slightly more recent date and thus could support a slightly more modest version of the authors' case. But it does not really indicate a substantially more recent age - like the attempt to move the wiggle match in SI Figure 6. The available dates on short-lived samples which should give contemporary ages are rather poor/ambiguous/inconsistent in several cases and offer less than good data. Some more would be good as would characterized wood/charcoal samples and then dates so we can distinguish in-built age from tree factors versus volcanic 14C.

6. The end result is that a relatively rather small group of other 14C dates (SI Figs. 4 and 5) - of which only some in fact indicate the date range preferred by the authors - end up as the key evidence and overturn a tight statistically good and replicated wiggle-match comprising many more high-quality 14C data. This is of course possible. The authors have made an interesting case in support and it is certainly worth considering - and is much better in their revised manuscript and materials. But nonetheless I see the evidential basis as in fact rather thin/ambiguous for the central claim of the re-dating of Taupo (by very much). The paper is more a call for more high-quality dating work to either disprove or prove their hypothesis. I would suggest rephrasing the paper a little to give it such a slant - this works to the benefit of the authors and the field.

Minor comments:

SHCal not sh

SI Fig.7 - there is no "0" year CE.

Overall? I find the paper interesting and the HYPOTHESIS proposed is intriguing and worth investigation. Thus I would support publication BUT ask/expect that the authors address the issues in my points 1-6. And as noted in 6 I would make the slant more clearly a hypothesis that is potentially important and now needs proper resolution (since the data to hand so far are not really definitive).

Responses to Reviewer #2

None of Reviewer #2's points disprove the existence of contamination or a younger date for the Taupo eruption, and hence potentially for other events worldwide. However, the reviewer does raise some valid questions on the absolute amount of contamination and age adjustment, with which we concur, and in light of which have made appropriate minor adjustments.

1. Any contamination affecting a tree could vary throughout its life, according to temporally variable and unconstrained magmatic CO₂ flux. The consequence is that ages within a wiggle match sequence could affect the height and location of individual peaks and troughs in the wiggle plot, making correlations less meaningful. We feel that the OxCal test is unreliable for that reason. Changes in the titre of old carbon laid down by the tree in each decade would stretch and contract the CRA sequence, accordion-style. We believe that the changes in the wiggle match calibrated chronology versus the actual passage of time shown by the ring sequence itself, with a final plateauing suggests strongly that such varying biases were present. Each date measurement could be moved independently by variation in the level of contamination, which could occur at decadal or lesser scales.

Such issues will of course also affect any attempted match at other points on the calibration curve. We thank the reviewer for reminding us that the potential problems in matching the original wiggle matches to the calibration curve could also affect the matching of that sequence to any other part of the curve. *We have therefore removed the figure of the alternative, later, match from the Supplementary Information and its call-out in the text.*

2. Our view is that there was continual, but variable, contamination of the ground water body, by gas being injected into it by basaltic magma (Barker et al. 2016) over the 250-year life of the tree in the lead up to the eruption sequence. We do not invoke a stable groundwater contamination, and hence *we have removed the 2nd wiggle match from supplementary material and paper (as discussed in point 1)*. We thank the reviewer for clarifying this in our minds.

3. The wiggle match trees are indeed near 60 km limit of contamination posited in the paper. However, the wiggle match tree site is northwest of Lake Taupo, adjacent to the general trend in groundwater movement in that direction, along the course of the Waikato River. Contamination of carbon samples for radiocarbon dating taken from along the course of the river has been suggested (Beavan-Athfield et al. 2001) and the groundwater and river are continuous. The contamination offset should therefore be present but not necessarily be large, in that area. *We have added a discussion (Page 4, lines 27-31, with the position of the wiggle match tree in relation to Lake Taupo and the Waikato River)*. The level of potential contamination in the wiggle match tree area is confirmed by the earlier wiggle match series, which gave an identical result for the date of tree death. We agree with the referee that an obvious further test of the contamination hypothesis would be a series of radiocarbon measurements and associated ¹³C/¹²C ratios on a tree from *outside* the posited contamination area, but which lived and died in the same time frame as the wiggle match tree. *We have added text to that effect (page 7, lines 16-20)*.

4. We have presented the calibrated probability distributions in the modified Fig. 1 (as requested). We agree that the error bars in the early measurements are wide – the focus at that

time was for sites near the volcano and the dating methodology could not generate higher precisions in the measurements – but as can be seen in Fig. 1 the bulk of the probability distributions for the calibrated dates on samples within 60 km of the vent lie before the wiggle match age. Most of the probability distributions for ages from beyond 60 km lie above the wiggle match age, i.e. the samples are younger than the wiggle match age. We feel we are justified therefore in using the median, as measure of central tendency in the data, in investigating the relationship between distance and age.

Our contention is obviously, therefore, that the contamination affected the close samples more than the more distant samples. We agree with the extra-60 km modelled age ranges of 249-337 CE (68.2%) and 206-388 CE (95.4%) as presented by the reviewer. However, the wiggle match age of 232 ± 4 CE is only just within the 95.4% calibrated date distribution, but it is **significantly outside** the 68.2% distribution and far from the middle of both. Taking into account the 68.2% distribution as above, we concur with the reviewer's point that on present evidence there is a more modest offset than in our original MS and have amended the text to that effect, while retaining the point that this is still likely to be a maximum age for the ignimbrite event. *We have now presented ranges and means for the different ways of combining the ages and discussed the implications (Page 9, lines 12-24).*

5. We agree with the referee that inbuilt age of wood and charcoal samples can be a serious problem with radiocarbon dating, particularly early in its use, when such matters were little understood. We feel, however, that it is unlikely that all the within-60 km wood-based ages were on inbuilt age wood and those outside 60 km were on short-life material, this would be an unlikely coincidence. We suggest that an *a posteriori* hypothesis of geographic array of wood and charcoal in-built age is implausible at best. *The potential problem of inbuilt age is now discussed in the text (Page 3 line 29 to Page 4 line 2).*

6. Our title indicates that we are offering evidence for geologic carbon contamination of materials used for radiocarbon dating of volcanic eruptions, when those materials are exposed to gases derived from underlying magma. We propose that the evidence we present supports the hypothesis that magmatic CO₂ contamination at unknown, but significant, levels has yielded anomalously old radiocarbon ages for samples from the volcanic edifice and its environs. *We now include the potential further test for this hypothesis (Page 7 lines 16-20).* For the Taupo First Millennium eruption (TFME), we suggest that the eruption occurred within a date range of at least a century but up to 200+ years younger than the presently-accepted wiggle match ages. Importantly, even a c. 100-year shift in the age for a prehistoric eruption such as the TFME has serious implications for correlations with both the climatic and ice core records.

References

- Barker SJ, Wilson CJ, Morgan DJ, Rowland JV. 2016. Rapid priming, accumulation, and recharge of magma driving recent eruptions at a hyperactive caldera volcano. *Geology*. 44:323-326.
- Beavan-Athfield NR, McFadgen BG, Sparks RJ. 2001. Environmental influences on dietary carbon and ¹⁴C ages in modern rats and other species. *Radiocarbon*. 43:7-14.

Reviewer #2 (Remarks to the Author):

This is a third revision. I do not return to matters covered in two previous reviews.

The authors have addressed the previous questions and concerns raised by this reviewer before, and, while one could go on debating various points, I regard this revised manuscript now as a good manuscript which is much improved and which makes/raises an important and relevant issue. Thus the manuscript deserves publication and attention - and hopefully it will lead to some new work to provide appropriate evidence from which to address/test some of the more challenging hypotheses proposed.

The major claim (magmatic CO₂ may potentially affect the ¹⁴C dating of some volcanic eruptions - depending on context as the authors outline in the last paras) and its specific case study (the 1st millennium AD Taupo eruption) are important to a range of fields and deserve a high profile publication. The paper will certainly lead to debate and should prompt new work to clarify and investigate its hypotheses - which are potentially important.

REVIEWERS' COMMENTS:

Reviewer #2 (Remarks to the Author):

This is a third revision. I do not return to matters covered in two previous reviews.

The authors have addressed the previous questions and concerns raised by this reviewer before, and, while one could go on debating various points, I regard this revised manuscript now as a good manuscript which is much improved and which makes/raises an important and relevant issue. Thus the manuscript deserves publication and attention - and hopefully it will lead to some new work to provide appropriate evidence from which to address/test some of the more challenging hypotheses proposed.

The major claim (magmatic CO₂ may potentially affect the ¹⁴C dating of some volcanic eruptions - depending on context as the authors outline in the last paras) and its specific case study (the 1st millennium AD Taupo eruption) are important to a range of fields and deserve a high profile publication. The paper will certainly lead to debate and should prompt new work to clarify and investigate its hypotheses - which are potentially important.

RESPONSE

As Reviewer #2 does not raise any new points for clarification or amendment, we have proceeded to revise the MS in accordance with the copy editing and suggestions. We thank Reviewer #2 for their most constructive comments and criticisms during the review process.